# A Systematic Measurement of Street Quality through Multi-Sourced Urban Data: A Human-Oriented Analysis

**DOI:** 10.3390/ijerph16101782

**Published:** 2019-05-20

**Authors:** Lingzhu Zhang, Yu Ye, Wenxin Zeng, Alain Chiaradia

**Affiliations:** 1Department of Urban Planning and Design, Faculty of Architecture, The University of Hong Kong, Hong Kong, China; zhanglz@hku.hk (L.Z.); wzeng04@connect.hku.hk (W.Z.); alainjfc@hku.hk (A.C.); 2College of Architecture and Urban Planning, Tongji University, Shanghai 200092, China

**Keywords:** systematic measurement, street quality, multi-sourced urban data, urban design, human-oriented, Shanghai

## Abstract

Many studies have been made on street quality, physical activity and public health. However, most studies so far have focused on only few features, such as street greenery or accessibility. These features fail to capture people’s holistic perceptions. The potential of fine grained, multi-sourced urban data creates new research avenues for addressing multi-feature, intangible, human-oriented issues related to the built environment. This study proposes a systematic, multi-factor quantitative approach for measuring street quality with the support of multi-sourced urban data taking Yangpu District in Shanghai as case study. This holistic approach combines typical and new urban data in order to measure street quality with a human-oriented perspective. This composite measure of street quality is based on the well-established 5Ds dimensions: Density, Diversity, Design, Destination accessibility and Distance to transit. They are combined as a collection of new urban data and research techniques, including location-based service (LBS) positioning data, points of interest (PoIs), elements and visual quality of street-view images extraction with supervised machine learning, and accessibility metrics using network science. According to these quantitative measurements from the five aspects, streets were classified into eight feature clusters and three types reflecting the value of street quality using a hierarchical clustering method. The classification was tested with experts. The analytical framework developed through this study contributes to human-oriented urban planning practices to further encourage physical activity and public health.

## 1. Introduction

### 1.1. Quality-Focused Studies on the Built Environment

In recent years, research focusing on the interaction between quality of life and the built environment has been extensively explored. Various research strands have examined how the design of the built environment could contribute to the establishment of more vibrant communities providing essential benefits to public health [1,2,3,4]. For example, Handy et al. [2] investigated the relationship between neighborhood characteristics and travel behavior. Ewing and Handy [4] proposed a conceptual framework by which physical characteristics of streets could be used to comprehensively measure the street environment and, ultimately, walking behavior. A large subset of studies have validated the impact of urban design on the quality of the built environment concerning themes such as human health [5,6], pedestrian activity [7,8,9,10] and travel behavior [11,12].

Planning authorities and research institutes are also increasingly encouraged to take up more quality-focused and human-oriented approaches to meet people’s demand for a lively and attractive built environment. Recently, Transport for London has turned its focus towards conducting studies on health-oriented built environments in order to create healthy streets eventually benefitting public health [13]. New York City and the Center for Active Design have collaborated to publish The Active Design Guidelines: Promoting Physical Activity and Health in Design, aiming to address obesity-related issues through a strategy of physical space design [14]. In Singapore, since 2010 the Future Cities Laboratory (FCL) has collaborated with the Urban Regeneration Authority (URA) to seek strategies to develop a sustainable city by applying science to design, exploring how the walkability of Singapore affects people’s behavior and route choices [15]. However, planners have been criticized for failing to address the concept of quality of life and measurements created by non-professionals for specific purposes have been called eccentric [16].

### 1.2. Streets as a Key Element of the Built Environment

Studies addressing street-quality from a more detailed perspective have recently been proposed. Evidence suggests that streets, as a key feature of the built environment, could efficiently encourage physical activities such as walking and cycling and significantly affect public health if high-quality designs were employed. Streets not only function as carriers to transport people but are also regarded as urban public spaces [17]. To provide an objective evaluation of street quality, quantitative approaches are being developed and introduced gradually. One such approach is to study the effects of network characteristics on people’s travel walking and route choice. A number of previous studies have indicated the relations between street-level physical accessibility measures and human behavior at different spatial scales, whether walking, cycling, driving, riding car, bus, metro or train [18,19,20,21,22]. An empirical study by Turner made a comparison between angular and block distance measures, demonstrating that both angular (most direct) and Euclidean (shortest) measures correlate well with the observed vehicular flow and thus suggesting that angular and Euclidean metrics could be combined to reflect genuinely shortest routes of the system [23]. More recently, a study examining the walking route choice data of pedestrians in Brisbane has claimed that pedestrians tend to minimize both the directional change and path length if they can [20]. In the case of five metro station areas in Shanghai, Zhuang and Song examined the influence of number of lanes together with network configuration on vehicular flow [24]. It was found that these two indicators explain 60% to 75% of vehicle movement volume variance. Sarkar’s research [25] has reported street-network configuration and street design factors such as urban green space, street connectivity (“betweenness”) and proximity to service destinations, all positively influence people’s walking behavior.

Moreover, eye-level street greenery as an essential street quality component has also been examined. Some studies have found that there is a strong, positive relation between eye-level street greenery and pedestrian behaviors that often contribute to public health [26]. Another study used street view pictures in conjunction with a machine-learning method, to assess the visual quality of hutongs in Beijing based on greenery, openness and enclosure. These 3 factors play a key role in urban vitality and have been closely associated with users’ willingness to stay [27]. Street quality also influences urban vitality and health from a broader perspective. Through an exploration of points of interest (PoIs) along the street network and human activity records using location-based services (LBS), urban vitality was evaluated, and a new urban residential project was determined to be dramatically less vibrant than an old urban area [28]. Using similar datasets, Ye et al. [29] used regression models to study the vitality of urban spaces in relationship to urban morphology and how design was implemented by using open data from a small catering business in Shenzhen. Through a quantitative analysis based on PoIs, it was established that two essential elements—density and typology—played key roles in determining urban vitality.

### 1.3. Challenges and Opportunities for Street Quality Measurement with the Help of New Urban Data

Historically, urbanists have debated the connection between street quality and social activities on streets. Lynch [30], Hedman [31], White [32] and Jacobs [33] all emphasized the value of physical characteristics for street life, suggesting that better perceptual street quality may create a stronger responsiveness to walking along the street. In recent years, quantitative approaches have been gradually introduced into this field [4,34]. However, the previous time-consuming manual data collection process may not satisfy the needs of future large-scale studies. Furthermore, the existing studies are of a wide variety but are limited in their ability to be broadly applied because of limitations in past techniques and data accessibility, especially with respect to human-oriented and perceptual-based street quality measurements. Still, the correlation between design elements affecting street quality and human walking behavior has been noted, as evidenced by the referenced research that explores this topic from many perspectives. However, as Handy et al. [2] have pointed out, there is a need for a more refined measurement method and more comprehensive data for analysis. This lack of effective techniques and data has led to the absence of an integrated measurement. Moreover, the existing top-down perspective in urban planning has been regarded as being poorly suited to measure the perceptual qualities of the urban environment. With the rapid development of new urban data, previously unmeasured elements could be assessed, thus opening potential new methods for measuring and evaluating intangible elements that are judged through perceptions, such as urban vitality and spatial qualities [35].

Rapid urbanization has created a need for refined urban design. Emerging high-resolution open data of road networks, PoIs and building base maps provide new opportunities for measuring neighborhood vibrancy at human-scale [36,37,38]. Additionally, the customization of Python and ArcGIS has accelerated mass data processing time. The series of emerging data and techniques provides not only possibilities to measure easily perceived but intangible data but also a new perspective from which to examine immaterial factors, thus facilitating the creation of a refined and efficient community. Algorithms based on convolutional neural networks like SegNet [39] and YOLO [40], can identify and extract physical elements from a serial sequence of images; furthermore, similar algorithms, like support vector machine (SVM) and random forest (RF), are beginning to be employed in the study of built environments to address complex, non-linear relationships [41,42].

Following this trend, the main objective of this study is to develop a systematic measurement method based on users’ sensibilities into a form of scientific street-quality assessment, aiming to achieve co-presentation of human-scale measurement with citywide analyses. Thus, a rapid but highly detailed evaluation of perceptual-based street quality can be provided to facilitate better understanding for urban planners, designers, policymakers and finally contribute to benefits of the public.

## 2. Literature Review

### 2.1. Attractive Built Environments: 3Ds and 5Ds

During the late 20th century, the idea of a mixed-use, compact development with a pedestrian-friendly built environment was suggested by many researchers. Dense blocks would encourage one to walk or cycle, not only reducing car-parking space but also creating opportunities for other functional development. Further, the distribution of various programs within a short distance from each other, facilitated by density, also promotes users’ willingness to walk [43]. Cervero and his colleagues studied the built environment and residents’ travel within the Bay Area in the United States. They concluded that the dimensions of density, diversity and design are the three most fundamental indicators for the evaluation of the built environment and are closely associated with user’s travel mode of choice and their odds of walking [44]. Based on earlier studies, Cervero and Ewing [45] updated the previous concept of 3Ds as 5Ds, adding the dimensions of destination accessibility and distance to transit and focusing on measuring the relationship between the built environment and travel behaviors in a more systematic manner.

Previous studies have been conducted based on Cevero’s 5D principles of Transit Oriented Development (TOD) to explain trip frequencies, travel mode choice, travel distance and overall vehicle miles travelled (VMT) [8,9,11]. The 5Ds framework has been recognized internationally by researchers and professionals for classifying and measuring attributes of the built environment with the aim of gaining a better understanding of physical activity and travel behaviors. Most recently, this framework was examined within the context of Asian cities. In an examination of Hong Kong, Lu et al. [26] used the 5Ds framework to study the relationship between households’ commuting mode choice and various dimensions of the built environment. Through an analysis of people’s commuting mode choices in Hong Kong, the study discovered that the factors of design and accessibility have a strong effect on human travel behaviors. However, density and diversity were not decisive under such a high-density urban condition. Thus, as mentioned by Lu et al., due to their compact urban morphology and distinct cultural background, the 5Ds framework by Cervero and Ewing must be adjusted in order to more properly fit Asian cities’ context.

### 2.2. Conceptual Framework: Measuring Human-Oriented Street Quality as the Combination of the 5Ds

As discussed previously, the development of technology and increased data access made it possible to quantify street-quality measurement. Expanded from Cervero and Ewing [45], this study adjusted the 5D variables in order to better understand and evaluate Shanghai’s built environment.

Density is a variable measured as “interest per an area” [45]. It has been recognized as one of the most essential and often-used built environment variables [46], as well as one of the two essential elements within the concept of urban vitality [47]. Many studies have examined the relationship between density and the physical environment, most commonly based on dwelling density and employee density, suggesting that density has associations with ridership, commute-mode choice and travel purposes [48,49,50,51]. With the progress and popularization of the Internet and the emergence of Information Communication Technology (ICT), it is common to see people using mobile internet LBS. This positioning data provides real-time user information with precise locations. Therefore, street vitality, which Lynch [52] holds as the primary criterion in the assessment of the quality of urban space, can be measured using LBS position data gathered over a long term and over a wide range. Also, compared to traditional measurement methodologies, using LBS data could identify fluctuations in urban vitality with regards to time differences [47].

Diversity is considered to be another important element of vitality [47]. Mixed land-use can encourage people to walk and to use public transportation, thus increasing opportunities for physical activities on the street. Diversity within this study is measured on the basis of PoIs through the Shannon-Wiener Index, which has been used in many built environment studies [53,54]. As Marcus suggested, a combination of diversity and accessibility, also known as urban planning capacity, has a greater impact than density when discerning urbanity [55].

Design includes the collective effect of multiple design factors, such as block size, crossings, street intersections, building setbacks and others [45]. This study used measurements based on perceived quality, allowing multiple attributes to be evaluated. The six key elements were categorized as street greenery, sky view, building frontage, pedestrian space, motorization and diversity reflected by other design elements, selected on the basis of the discussion of a series of classical urban design theories. There is a constant effort from urbanists focusing on key design elements and perceived street quality. A systematic review has been made to go through related design theories from Jacobs [33] to Trancik [56] to Katz [57] and Montgomery [58]. Only the elements with clear operational definitions are selected. Specifically, street greenery would lead to beautiful screens and emotional pleasure that can help to create lively streets. The sky view directly affects the enclosure of streetscape and the building frontage would encourage potential interactions between buildings and streets. Both of these two elements would help to the increase of perceived quality on streets, which have been well mentioned by many urbanists and empirically tested by Gehl [59]. Wider pedestrian space would bring positive effects on pedestrian activities and potential social interactions and then increase street quality. In turn, motorization, that is, the width of motorways, would be negative effects. Moreover, diversity would increase the willingness to walk and cycle on streets and finally contributes to the perceived quality as well. The traditional study processes that explore the relationship between visual quality, space perception and corresponding design elements are tedious and applicable only to small-scale studies. However, the eye-level street view images derived from Google Street View or Baidu Total View are sufficient for the measurement of a street’s urban quality with a more efficient process and an even better quality. In previous studies, an evaluation model with an artificial neural network was developed, which could quantitatively measure the perceptual-based spatial quality of urban streets using street view images and machine learning algorithms in a large-scale context [35].

Destination accessibility was recognized as one of the most fundamental factors for urban physical activity. This study follows the definition of accessibility given by Shimbel [60], suggesting that more accessible areas contain better opportunities for interaction and thus growth potential. This involves the cost distance from one particular point to another within the road network, suggesting the degree of difficulty in satisfying social activities within the area. The representation of urban spatial networks based on graph theory provides various possibilities to evaluate accessibility in general. For instance, spatial Design Network Analysis [61,62] (sDNA)—an urban network analysis technique developed by Cardiff University—measures street accessibility and flow potential by describing streets and junctions as nodes and links.

Distance to transit is usually measured as the shortest network distance from an origin to a nearby transit stop [45]. Public transit use is thought to encourage more physical activities since it often requires people to travel a certain distance via walking or cycling to reach both their starting point and destinations [63,64]. A study in Shanghai found that 60% of commercial land use are located on the roads with the 30% highest level of flow potential from micro to macro radii and 76% of metro stations and 65% of bus stops are located on the roads with 30% highest level of flow potential at large radii, indicating a coupling multiplier effect between land-use, transit stops and network spatial accessibility [22]. Transit facilities within walking distance in a community, therefore, promote not only a convenience for users but also the use of public space. The walking distance between PoIs to the closest transit stop was measured in this study in order to evaluate the convenience factors within the area.

In summary, a range of studies has investigated the relationship between 5D values and built environment, while few of them have discussed this 5Ds framework as a whole. There is a need to analyze street quality and better understand how it promotes daily activities from the point of view of uses.

The main contributions of this study are:
Use of the 5Ds framework discussed above, considering the distinctive physical and cultural features of the Asian cities context. This study strives to re-integrate these five variables and examine them in the Shanghai context with the primary objective of providing a comprehensive evaluation framework.Use higher data resolution to measure street quality and their relation to physical activities from people’s daily behaviors, portraying a more human-oriented approach.Considering that intersection density cannot fully describe street layout configuration and the relationship between part and whole for pedestrian and the serial view experience of the pedestrian, this paper uses a description of street layout and network science to present a more realistic pedestrian path choice routing analysis.


To the knowledge of the authors, this is the first time that a study shows a possibility of measuring street quality with a more systematic and human-oriented way, providing a comprehensive evaluation framework that is innovative and meaningful for future studies.

## 3. Materials and Methods

### 3.1. Analytical Framework

This study involved four major phases: data collection, element extraction, evaluation and systematic measurement (Figure 1). First, LBS positioning data and data concerning PoIs, street-view images (SVIs) and the street network of Yangpu District were collected. Second, five key variables affecting street quality were extracted from the dataset. Assessment of density was through the concentration of LBS data, diversity was calculated using the entropy of PoIs, design was defined as the visual quality of streets using six key design elements extracted from SVIs through SegNet, accessibility was assessed through betweenness centrality (through-movement potential) of each street link and distance to transit was measured by the walking distance to the closest metro station exits and bus stops. The evaluation phase contained two steps: First, a hierarchical cluster analysis was performed to group data instances into a tree of clusters; second, radar charts were used to display multiple quantitative variables for each cluster and to classify street quality into three types. Thus, a systematic measurement of street quality was developed to provide designer with the possibility of a deeper understanding of existing street quality, contributing to better planning management and street design.

### 3.2. The Study Area: Shanghai Yangpu District

Our analysis focuses on the Yangpu District in Shanghai, the area northeast of downtown Shanghai bordering the Huangpu River on its east and south side. It is predominantly composed of residential communities and industrial facilities with a population of 1.3 million as of 2015 and a land area of 60.61 km^2^ (Figure 2). By the end of 2018, Yangpu district had 4 metro lines, 23 stations, with a service coverage (800 m buffer area) ratio of 54%, substantial enough to deploy a TOD based 5D framework analysis.

As an old industrial district, Yangpu District had about 12.8 km^2^ of industrial land at the end of 1998, accounting for 21.3% of its total area. Since the early 21st century, this District has increasingly developed away from Industrial Yangpu (labor-intensive industry) towards Knowledge Yangpu (commerce and high technology) and the area around Wujiaochang boasts one of the top 10 shopping areas in Shanghai. Currently, a special work plan (2018–2020) titled Neighborhood Beautification [65] with the goal of improving street quality and building an enjoyable community in Yangpu District is ongoing. From the city perspective, promoting community quality through refined management seems urgent for better understanding and promoting better place-making.

### 3.3. Measuring Street Quality via Five Dimensions

#### 3.3.1. Density

Data from increasingly accurate GPS positioning using LBS are recognized as an appropriate quantitative measurement of walking, relaxing and other activities that take place on streets. This paper employs the intensity of human activities, based on crowd tracking, in order to measure street vitality. Real-time population locations from Tencent location-based data (https://heat.qq.com) at different times (7 a.m., 11 a.m., 3 p.m., 7 p.m. and 10 p.m.) on one weekday and one weekend were obtained to reflect the spatial distribution of the real population. A total of 151,264 positioning points was collected on a weekday and 146,464 on a weekend (Figure 3). 

Thus, an all-day population density was calculated using the average data for different times and weighting 5:2 for weekdays versus weekends. Kernel density estimation in ArcGIS was adopted to examine the spatial clustering of population locations. As shown in Figure 4, red areas on the map represent clusters identified as high-density hotspots. The next step was to spatially join the data with street networks to show the density values at street level and to further make the results comparable with the other four factors.

#### 3.3.2. Design

As stated above, the design dimension is defined as the perceived quality of streetscapes, which is intangible and hard to measure. Nevertheless, the recent emergence of SVIs and machine-learning algorithms have provided solutions for this issue. Pioneering studies have evaluated the perceived safety of urban areas based on their appearance [66], as well as the facade continuity of buildings [67]. Herein, the perceived quality of streetscapes is measured via a supervised machine learning approach [68].

First, six key spatial elements of streets, including sky view, greenery, building facades, pedestrian space, motorway space and street furniture and facilities were selected from the classic urban design literature [69,70,71,72,73]. These six elements were measured through the combined application of SVIs and SegNet. The SVIs used in this study were collected through Baidu Maps API and Python in April and May 2017 (Figure 5). The 38,729 sample sites were selected at 20 m intervals according to Gehl’s [74] theory of urban perceptions. Every site contains two street view images with both front and back views. Specifically, the SVIs were requested in an HTTP URL form using the Baidu Maps API [75]. By defining the URL parameters sent through a standard HTTP request, users can obtain a static image from any direction and viewing angle. SegNet, an advanced deep convolutional neural network architecture that maps each image pixel into semantics, was used to extract the key spatial elements of SVIs. For SVIs, a global accuracy of 90.4% can be achieved for a total of 12 classes, with the accuracy being even higher for the classes of buildings, sky, cars and roads [39]. The schematic architecture of SegNet is shown below (Figure 6). It has an encoder network and a corresponding decoder network, followed by a final pixel-wise classification layer. A series of encoder and decoder network composes of a deep convolutional architecture to achieve a soft-max classifier for pixel-wise classification. The SegNet applied herein is a pre-trained model provided by researchers from the University of Cambridge. We did not re-train it for this specific context as an empirical study utilizing this algorithm in Chinese cities performed well [76].

As shown in Figure 7, SegNet works well for extracting key spatial elements in the case area. Thereafter, perceived quality was evaluated on the basis of key spatial elements through an artificial neural network (ANN). Specifically, a large quantity of pairwise comparisons of representative images were collected via a Java-based program (Figure 8). One key question to answer was: which side looks better? Overall, 50,000 comparisons were made by ten experts with professional backgrounds and a familiarity with Shanghai. The results collected from the pairwise comparisons were then converted into scores through the Elo rating system [77], a widely used algorithm for comparing players’ capacity in one-versus-one games. The initial score for every representative image was set as 1000 and then the new results were computed through the Elo rating system for many rounds until the results become stable. The final scores are mainly distributed between 800–1800.

Thereafter, the ANN was applied to train an evaluation model utilizing the representative images containing a series of measured design elements and computed scores from the experts’ preferences (Figure 9). Preliminary observations showed that there was no clear linear relationship between the perceived quality and these key spatial elements. That is easy to understand. For instance, the increasing of street greenery within a certain threshold would bring direct positive effects on perceived streetscape quality. However, too much greenery over the threshold might bring negative effects. The decrease of a sky view means the increasing of street enclosure. Positive effects would be brought if this kind of enclosure is achieved by an appropriate mixture of greenery, building frontage and diversity facilities. Nevertheless, a very high enclosure caused by pure building frontage might lead to negative perception and depression. Therefore, linear regression cannot address this complex and interacted relationship appropriately.

Many machine learning algorithms, including ANN, Decision tree and Random Forest, have been tested. The ANN was finally selected as it fitted well in complex non-linear relationships and performed a higher accuracy compared with other algorithms. It is good at “learning” because of its consideration of examples to perform specific tasks: for example, using evaluated samples of street scores to rate the remaining ones [78]. Different combinations of hidden layers and the number of neurons in this ANN were tested with the 10-fold cross-validating approach, which concluded that one hidden layer with eight neurons performed the best estimation. The further increasing of hidden layers and neuron numbers would lead to overfitting. After the training of the ANN-based evaluation model, we then applied it into the entire site to achieve a large-scale, high-resolution analysis. The evaluated perceived quality satisfied the verification process of comparison to common understandings collected by local urban designers, thereby showing acceptable accuracy in judging the design quality of streetscapes.

#### 3.3.3. Diversity

With the help of Python and AutoNavi’s Map API, 85,778 PoIs were collected from within the case area to compute diversity (Figure 10). AutoNavi is one of the largest map service providers in China, providing accurate, geo-referenced data on the built environment and related urban facilities. These collected PoIs have been marked as more than one hundred kinds of urban functions, for example, restaurants, shops, hospitals. The issue of diversity, an elusive feature, was difficult to measure. As discussed by Gagster [79], it is a term that is hard to define precisely but everyone can experience it. With assistance from new urban data and GIS tools, we herein attempted to propose a quantitative measurement of diversity.

In order to achieve a human-oriented measurement of diversity, we did not employ the traditional land-use categories based on functions in plots or street blocks. Rather, we attempted to develop a more fine-scale approach, integrating the total quantity of urban facilities and the diversity index of urban functions represented by these facilities. Through this, diversity was measured in two steps: (1) identifying the total PoIs within a walkable distance of a street segment and (2) measuring the diversity via the Shannon-Wiener index [80]. “Walking distance” in people’s daily lives is set as 1000 m according to an empirical study of living convenience in China [81]. We choose to use the Shannon-Wiener index, originally developed in ecology, to measure diversity because this index shows a good performance in urban planning and management [53]. The PoIs were classified into four functional categories as working, commercial service, public service and transportation. The working category includes office buildings, institutions and so forth.; the commercial category includes shops, restaurants, grocery stores and so forth.; the public service category includes schools, universities, hospitals, urban parks; and the transportation category includes metro stations, bus stops, bicycle parking piles and so forth. Empirical studies in many cities show that a lively and attractive place often contains a high mix of these four main functions [35,36].

The formula is as follows:
(1)$$D_{i}=N_{i}\times SW_{i}$$
(2)$$SW_{i}=-\sum_{i=1}^{R}p_{i}*\ln p_{i}$$
where *D_i_* represents the final diversity value of a street segment *i*, $N_{i}$ represents the sum number of PoIs within the walking distance buffer of street segment *i* and $SW_{i}$ represents the Shannon-Wiener index among different urban functions within the pedestrian distance. *p_i_* is the proportion of urban facilities belonging to the *i*th type of functional categories and *R* is the total number of main functional categories, which is 4 in our analysis.

#### 3.3.4. Measuring Accessibility

Street-level physical accessibility has been used to explain the impact of urban morphology on walking and physical activity [25]. In our study, a street path-center line map extracted from route data in Baidu Maps was employed as the base street network database. The measure of betweenness centrality, which represents the frequency of each link *x* of the shortest and most direct path between each pair of other links *y* and *z* within the radius, was used herein [82]. It reflects the through-movement potential of each street link that could be selected by pedestrians or drivers. Betweenness centrality is measured by:
(3)$$betweenness\left(x\right)=\sum_{yz}n_{yz}^{x}$$
where $n_{yz}^{x}$ be 1 if *x* lies on the shortest path from *y* to *z* and 0 if it does not.

sDNA was used to calculate betweenness centrality on the network with a user-defined radius with different metrics—Euclidean (shortest path), angular (most direct path), topological (least turn path) and hybrid (both shortest and most direct path). As travel budget, a 600 m radius was used as a comfortable walking distance in Calthorpe’s TOD theory [83] and has been shown to be applicable to walking behavior in Shanghai [22]. Therefore, hybrid betweenness value at a catchment radius of 600 m was taken as an indicator of the degree of pedestrian through-traffic.

The hybrid metric is defined as:
(4)distanceforlink=a×ang+(1−a)×euc
(5)distancefornode=a×ang


Cooper et al. noted that for Angular-Euclidean hybrid metric, 0.25 ≤ a ≤ 0.5 gave good results [84]. It has been reported that a calibration of half angular–half Euclidean metric gives a stable result in terms of interpreting pedestrian movement [85]. Therefore, a default hybrid metric which combines 50% angular and 50% Euclidean metrics has been adopted in this study, that is, a = 0.5.

#### 3.3.5. Distance to Transit

On the basis of the large amount of collected PoIs, the distance to transit could be computed as well. We measured the distance to transit as the reciprocal of the shortest network distance from the midpoint of street segment to the nearest transit node within the radius of 1 km (15 min walking distance), that is, metro station or bus stop. The network distance herein is the length of the shortest geographical path between these two locations along the network.

## 4. Results

### 4.1. Quantitatively Measuring Street Quality

The descriptive statistics of individual variables are shown as box plots in Figure 11. The line across the box represents the median, the x in the box represents the mean, whereas the bottom and top of the box show the locations of the first (Q1) and third quartiles (Q3). Box plots of density, diversity and accessibility are comparatively short, indicating that overall streets have a high level of agreement with each other. Box plots of the other two variables, design and distance to transit show a large variation in values; the long upper whiskers suggest that streets are varied amongst the higher quality and further distance to transit stops.

Box plots illustrate that the measures are highly skewed. For comparison, each variable was ranked and then divided into five equal quintiles accordingly. The lowest quintile was given a score of 1 and the highest quintile was given a score of 5. Figure 12 shows the score of five built environment variables for measuring street quality at street level in Yangpu District.

Figure 12a shows the LBS population density of streets from the perspective of pedestrian networks and population density, where warm colors represent high values and cool colors indicate low values. According to the image, higher values are in the central areas of the district, such as the Wujiaochang, Siping Road and Kongjiang Road communities (see sub-districts in Figure 12f). The case of the Dinghai Road community is particularly notable, as it shows high values despite its edge location within the district. Because of many students from the University of Shanghai for Science and Technology makes this area popular.

Compared with Figure 12d, a high concordance between LBS population density and degree of accessibility can be noticed. For example, the above-mentioned areas have a denser road network and more extensive LBS data use, explaining why pedestrians more frequently choose to use this area in their daily walking behavior. On the contrary, the streets in the Xinjiangwan community and the riverside area showed not only less LBS density but also lower walkability quotient, potentially due to the following two reasons: First, the population density of the Xinjiangwan community is recorded as the lowest within Yangpu District [86] and second, the 15.5 km-long riverside region in Yangpu District occupies the main part of Yangpu’s old industrial area; thereby negatively affecting the distribution of density.

The diversity of the Xinjiangwan community and riverside region is also low for similar reasons (Figure 12c) and single-purpose (single-function) land use has a significant negative impact on street diversity. Moreover, the Wujiaochang community, as a sub-center of Shanghai, has concentrated on commercial and office development due to its single-purpose land use, resulting in the current low-diversity situation. As shown in Figure 12b, district-and neighborhood-level urban streets, especially streets around universities, for example, Fudan University, Tongji University and the University of Shanghai for Science and Technology, have high design scores. Meanwhile, main roads have relatively low scores in the design variable because of their transport functions. The design quality along the Huangpu River also needs to be improved according to the measurement. The levels of transit proximity to transit stops are illustrated with colors ranging from dark blue to light blue (Figure 12e).

### 4.2. Hierarchical Cluster Analysis

A hierarchical cluster analysis, which is a multivariate statistical method for grouping cases according to the similarity of their characteristics, was performed to classify the data. In this study, we classified streets using all five built environment variables. The dendrogram shown in Figure 13 summarizes the clustering process and reveals seven clusters of streets and one outlier (cluster 8). This cut-off number was chosen in order to obtain a small number of representative clusters. We then focus on the seven clusters to see similarities and differences between them.

The balance of street quality scores for each cluster can be presented using radar charts that display multiple quantitative variables, allowing for visual comparison (Figure 14). From the radar charts, we can see that the main features of clusters can be categorized into three types:

Type A—high quality: The radar charts of clusters 1, 2 and 3 are similar in shape but differ in their radial diagram and variables. Three indicators for streets in these clusters have excellent scores (>3.5) (Cluster 1: Density, design and distance to transit; Clusters 2 & 3: Density, diversity and distance to transit). Both density and distance to transit scored high on streets in these groups, while none or only one of the other variables scored low (<2), indicating that the scores of the five variables were relatively balanced. According to the definition of 5Ds, high density means more activities take place, distance to transit indicates high global accessibility due to shorter distances to transit stops.

Type B—medium quality: Variables in cluster 4 are similar to clusters 1–3 in terms of shape. Unlike Type A. Density and distance to transit for streets in this group scored high, the other three variables had poor scores.

Type C—poor quality: clusters 5, 6 and 7, essentially consisted of the low-quality streets as more than three variables received poor scores. It is noticeable that Density and distance to transit are both low scoring variables in these three clusters, indicating fewer activities and global accessibility leading to lower overall street quality.

The results corroborate the Jacobean’s hypothesis [33], the more eyes on the streets the better quality they are and vice versa.

Table 1 presents specific examples of individual types from the study area. 558 streets were concluded of being high quality (Type A, 45%) in our analysis and 168 streets were concluded of being medium quality (Type B, 14%). The rest 505 streets were concluded of being poor quality (Type C, 41%). As can be seen from the distribution of the 505 poor-quality streets (Table 1—Type C), most of the lowest-scoring streets are located along the edge of Yangpu District on the south-east of the district and in Xinjiangwan Community on the north-west of the district. Conversely, the streets in clusters 1–3 are largely located in the central area (Table 1—Type A). Streets in cluster 4 are considered to have the highest potential in terms of quality improvement due to its high pedestrian activities and transit proximity.

### 4.3. Validating the Analytical Results

The difficulty in validating these results is that the spatial quality itself is a normative value. As is a priori in urban design, the measured spatial quality of streets is intangible. In other words, it is an abstract of experienced planners and designers’ preferences in place-making. Therefore, it is challenging to produce a solid, standard reference against which to run a verification. In this study, we applied two alternative approaches.

The first approach was to compare our results with references of positive examples presented in Shanghai Street Design Guidelines [87]. According to the common sense of urban planners and designers in Shanghai, the examples mentioned in the guidelines can be regarded as idealistic scenarios. All four of the good examples in Yangpu District mentioned in the Guidelines (Daxue Road, Sujiatun Road, Fushun Road and Zhengtong Road) were assessed as Type A in our analysis. Therefore, street quality can be considered to have been measured accurately despite the small sample size.

The second approach was to compare the evaluated street quality with a professional panel of ten urban planning and design experts familiar with the study area. Each expert was asked to evaluate six randomly selected streets each as being of good, medium or poor quality according to the 5Ds framework (see Appendix A). Experts marked 16 streets as good quality, 15of which were within Type A and one of which was in Type C, indicating the match between the experts’ opinion to the computer evaluation reached 93%. Twenty-six streets were categorized as being of medium quality by the experts, of which 12 were assessed as Type B in our analysis, 9 were identified as Type A and 5 were identified as Type C. Most of the poor-quality streets selected by experts (17 of 18) were identified as having the potential for improvement (Type B or C). This exercise proved that there was a high correlation between our analytical results and the general understanding of urban designers.

Considering the degree of validity, the results were also evaluated using Kappa Statistics, which measure the inter-rater reliability of the agreement between the evaluations of “machine” and “experts.” Cohen’s kappa correlation coefficient and weighted kappa values (using STATA) were calculated. There was an overall kappa of 0.59 (95% confidence interval (CI)), with a weighted kappa of 0.65 (95% CI) and agreement of 84% (Table 2), indicating a substantial level of agreement overall.

## 5. Discussion

### 5.1. Measuring the Unmeasurable: Evaluating Street Quality with Large-Scale and High-Resolution Data

Though the approach of quantitative measurement of street quality is not a new focus within urban planning and design, it had been difficult so far to apply it in broader contexts due to the limitations in technology and data. This paper develops a more comprehensive and systematic approach of measuring the unmeasurable, providing researchers with a more convenient and effective evaluation process. With the support of new data and innovative technology, high-resolution, large-scale analyses can be realized. Additionally, this methodology can be adapted within different fields because of its general applicability and the easy accessibility of relevant datasets as researchers forge a new science of cities.

### 5.2. Multi-Factor Measurement Combining Classical Thinking and New Analytical Tools

Various measurement tools have been used to evaluate the quality of the walking environment in recent years. However, these evaluations have so far been based on individual features which are not effective in capturing people’s overall perceptions of street environment [88]. In light of this limitation, this study re-interprets the classic 5Ds framework, an effective tool for measuring attributes of the built environment based on people’s daily lives and provided insights on spatial quality measurement using new urban dataset and the latest developments in machine learning technology. Summarizing five key elements for environmental quality measurement, this paper conducted a scientific and quantitative measurement of each street (N = 1231), allowing urban design evaluation to be structured based multiples perspectives summarized in easy to read radar charts. For instance, the Knowledge & Innovation Community located within the Wujiaochang community was scored as having a good overall quality (Figure 12f). However, areas for improvement, including pedestrian network density and connectivity, could be recommended based on the low accessibility score of certain streets.

### 5.3. Implications for Urban Policy and Design Practices from the Human-Oriented Perspective

This study’s findings also have direct policy and design implications. In the context of rapid urban development, the traditional top-down approach to urban planning might no longer be able to fulfill the needs of human-oriented city zoning and refined urban design. Systematic measurement with large-scale and high-resolution data from multiple perspectives can promote more human-oriented planning practices within an urban acupuncture approach to community beautification. First, this new measurement framework could provide an overall street evaluation through the analysis of multi-sourced urban data. Second, the scoring of individual elements highlights a clear direction for cities to optimize environment quality. Finally, the public’s street evaluations could be recorded with the support of a data platform, therefore creating the possibility of public participation. As illustrated in the paper, the proportion of the three street-quality levels within Yangpu District are 45%, 14% and 41%, respectively. Notably, most of the low-quality streets are located in the Xinjiangwan community and the others are positioned in the old industrial district along the Yangpu River (Figure 12f). This aligns precisely with the transportation and spatial planning guidance that the government announced. On one hand, the second phase of Metro Line 10 is under construction and will be extended from Xinjiangwan Station, passing through the Huangpu River and reaching the Pudong District directly; On the other hand, the 15.5 km-long development site along Riverbank was clearly singled out in the “13th Five Year Plan of both sides of Huangpu River” [89]. When future urban regeneration projects are implemented on the Huangpu Riverbank, the street quality of the whole riverside area will be greatly improved, which may contribute to an increase in physical activity.

### 5.4. Limitations and Next Steps

Several limitations must be addressed within this study. First, the sampling of LBS positioning data is selective and might not be representative and thus cannot compete with cellphone signaling data. Further efforts could attempt to correct LBS data based on cell-phone signaling data. Second, current analysis only uses the sum amount of behavior records as the reflection of density. The temporal changing of behavior density is an important issue deserves a systematic analysis in our following studies. Moreover, the multi-source data are from different time periods. Consistent and long-term collection of data is required in order to have more accurate information within the same time period. In addition, differences might exist between the preferences of the public and those of experts. Therefore, further endeavors are needed to collect large-scale data from local residents, workers and visitors. This could possibly narrow the gap between expert scoring and public experience. Finally, integrating machine learning algorithms into the comprehensive measurement of street quality is worth further investigation and cross-validation.

## 6. Conclusions

The perceptual-based quality of urban streets has been regarded as an important public good and there is an increasing interest in environmental studies related to public health. This study developed a systematic street quality measurement by reinterpreting the classic 5Ds framework and incorporating newly-emerged urban data. Combined with LBS positioning data, PoIs, eye-level SVIs, machine learning and street network analysis, the measurement of streetscapes can be quantitatively achieved from the perspective of people’s daily behavior. The development of analytical approach and understandings in this direction may contribute to efficient evaluation on street quality, further assisting appropriate planning interventions and encouraging physical activities and public health. We expect this innovative and human-oriented approach to potentially supplement and improve street renewal projects by promoting engagement between current research interest trends and new data and technology.

## Figures and Tables

**Figure 1 ijerph-16-01782-f001:**
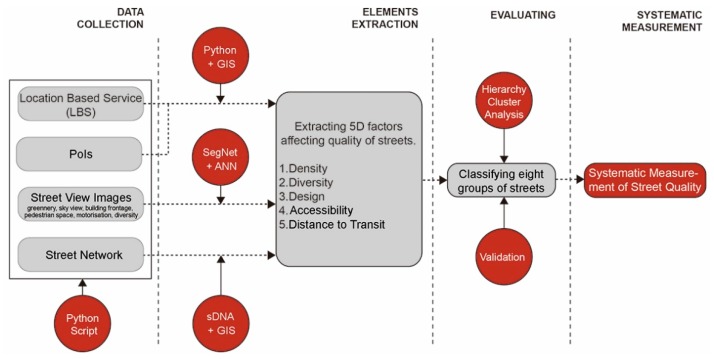
Analytic framework.

**Figure 2 ijerph-16-01782-f002:**
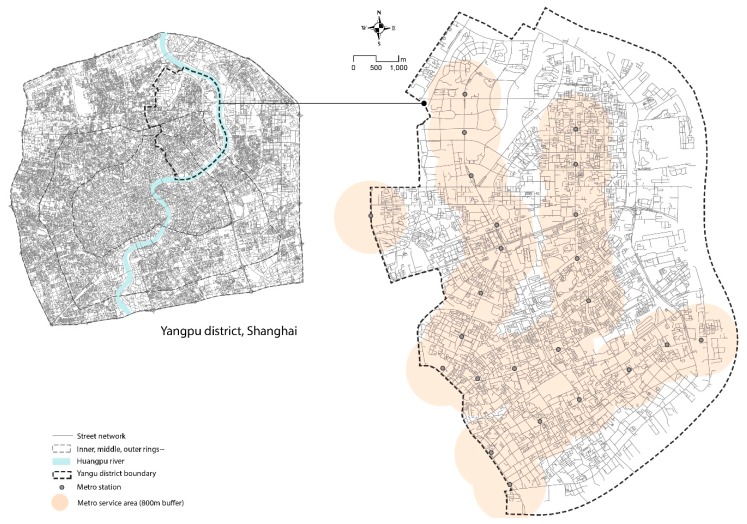
The study area and metro service area.

**Figure 3 ijerph-16-01782-f003:**
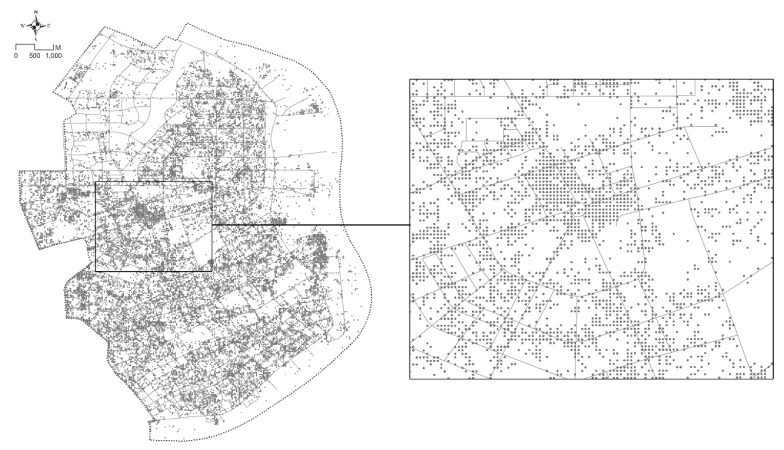
Example of location-based service (LBS) data on a weekend (10 p.m.) from Tencent (collected on 16th December 2018).

**Figure 4 ijerph-16-01782-f004:**
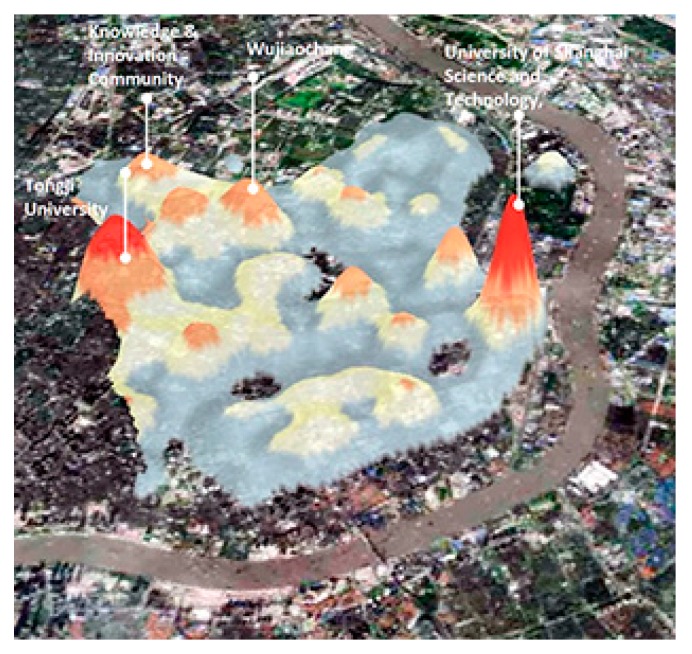
Kernel density hotspot map showing average population density of Yangpu District.

**Figure 5 ijerph-16-01782-f005:**
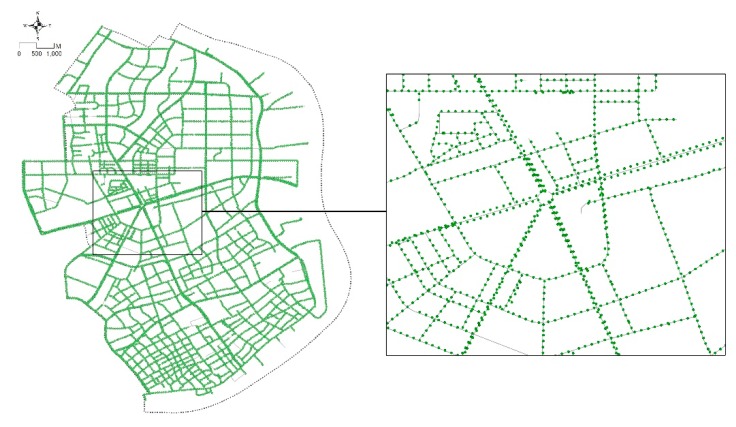
Distribution of sampling points for capturing street-view images (SVIs) of Yangpu District.

**Figure 6 ijerph-16-01782-f006:**
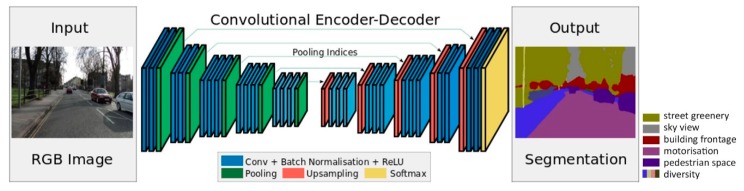
A schematic architecture of SegNet [39].

**Figure 7 ijerph-16-01782-f007:**
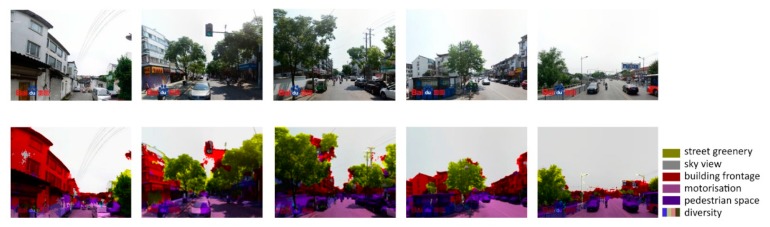
Applying SegNet to extract key spatial elements.

**Figure 8 ijerph-16-01782-f008:**
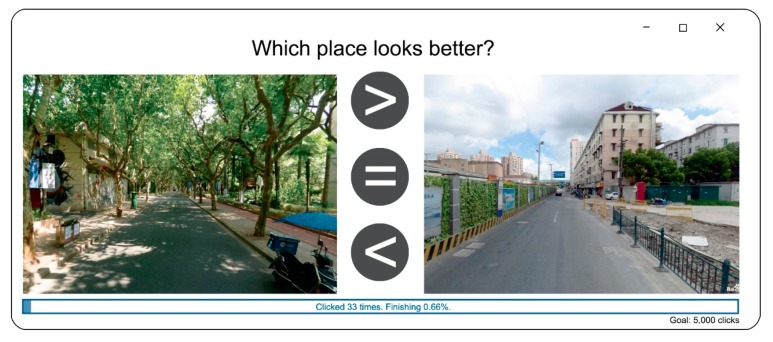
Comparison on representative images via a Java-based program.

**Figure 9 ijerph-16-01782-f009:**
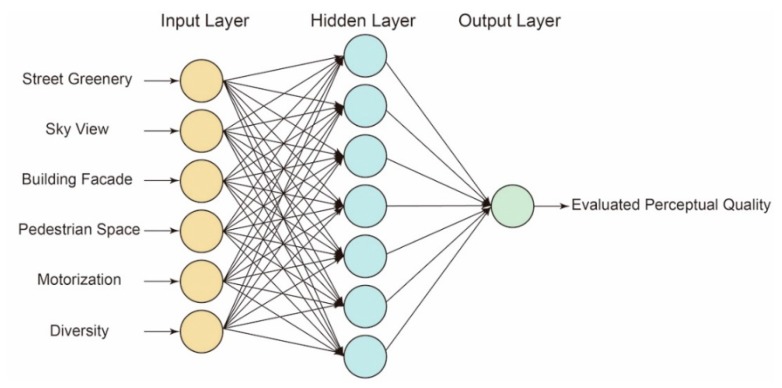
Artificial neural networks (ANN) implementation of the analysis of key spatial elements.

**Figure 10 ijerph-16-01782-f010:**
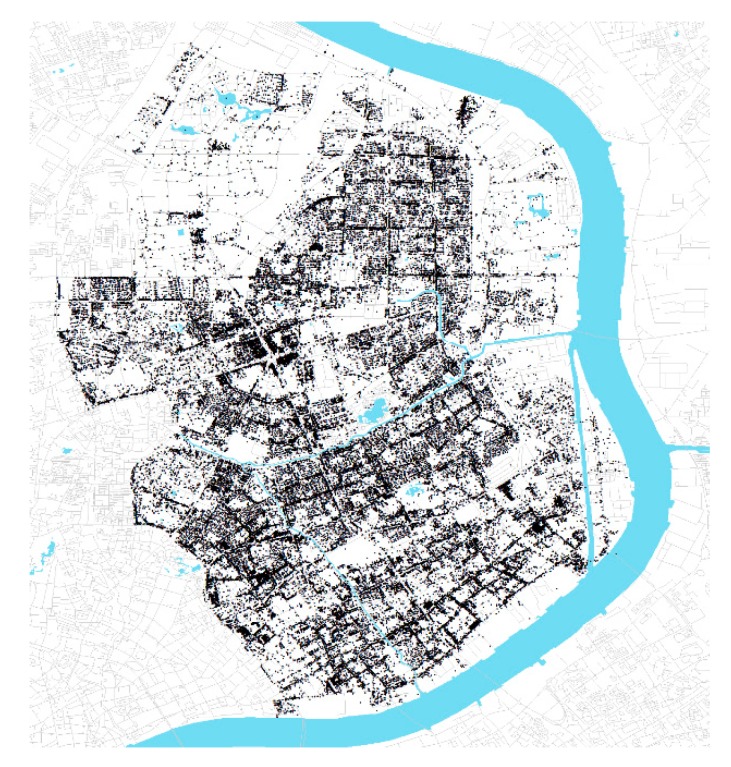
Distribution of points of interest (POIs) of Yangpu District.

**Figure 11 ijerph-16-01782-f011:**
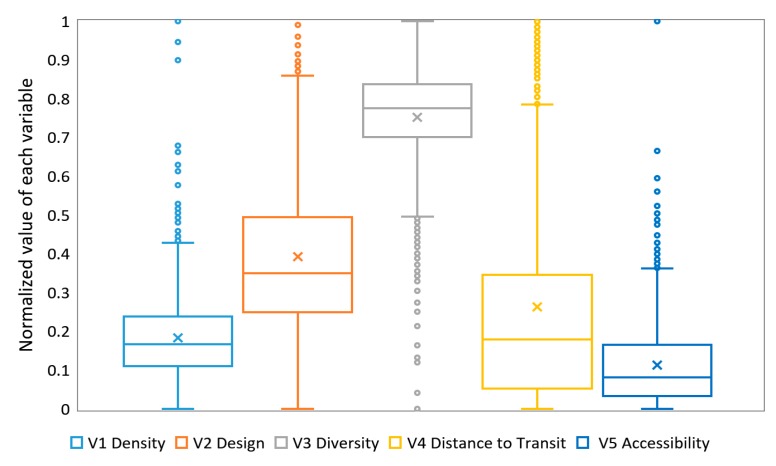
Box plots for density, design, diversity, distance to transit and accessibility values (normalized). The whiskers extend from the ends of the box to the minimum value and maximum value. Individual points with values exceed a distance of 1.5 times the inter-quartile range (IQR = Q3 − Q1) below Q1 or above Q3 are plotted with circles.

**Figure 12 ijerph-16-01782-f012:**
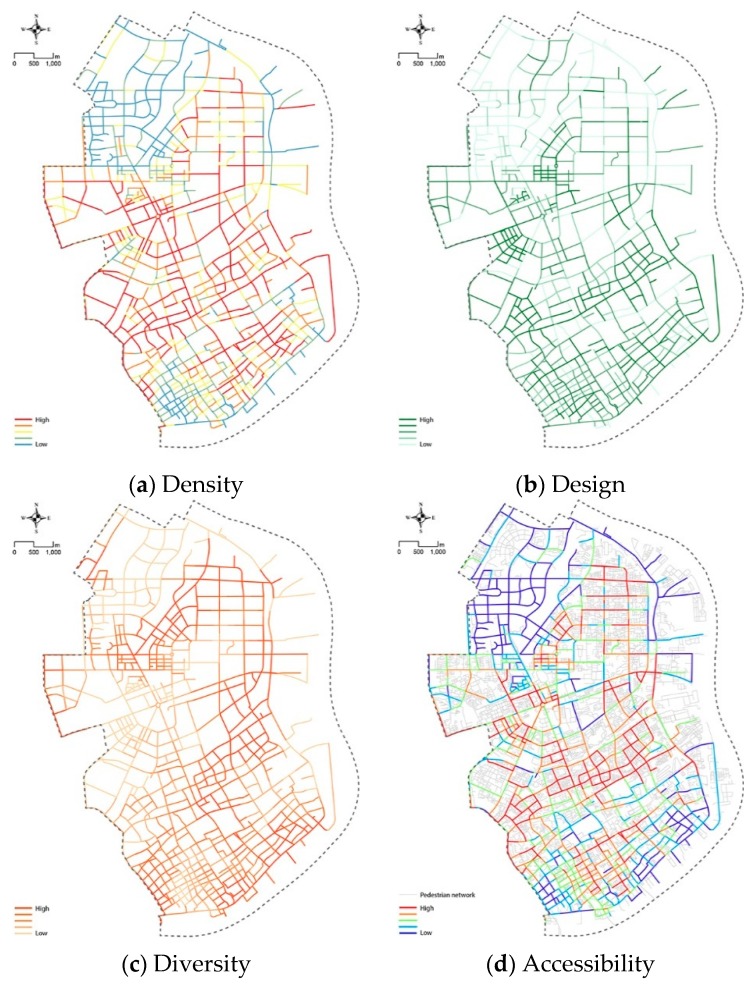
Measuring built environment elements of street quality.

**Figure 13 ijerph-16-01782-f013:**
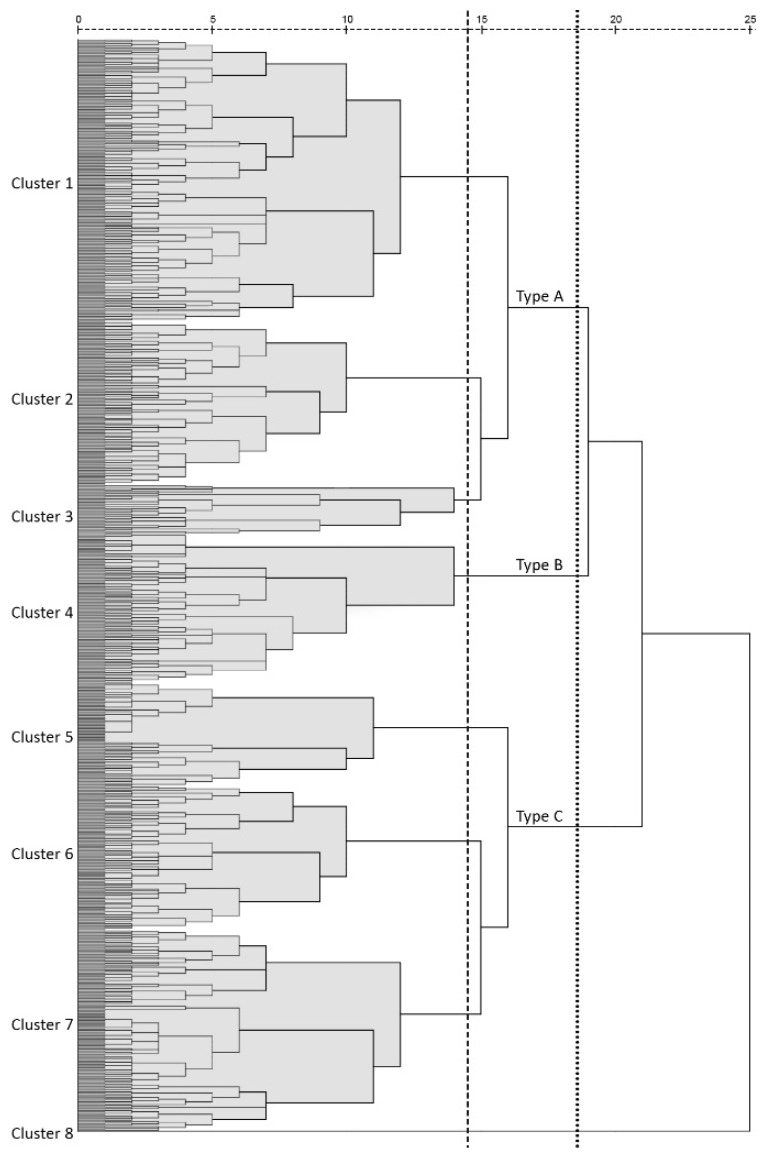
Dendrogram using Average Linkage (Between Groups) corresponding to the cluster analysis based on five built-environment variables. The vertical lines indicated the normalized distance at which the eight clusters (dashed) and three types (dotted) confirmed.

**Figure 14 ijerph-16-01782-f014:**
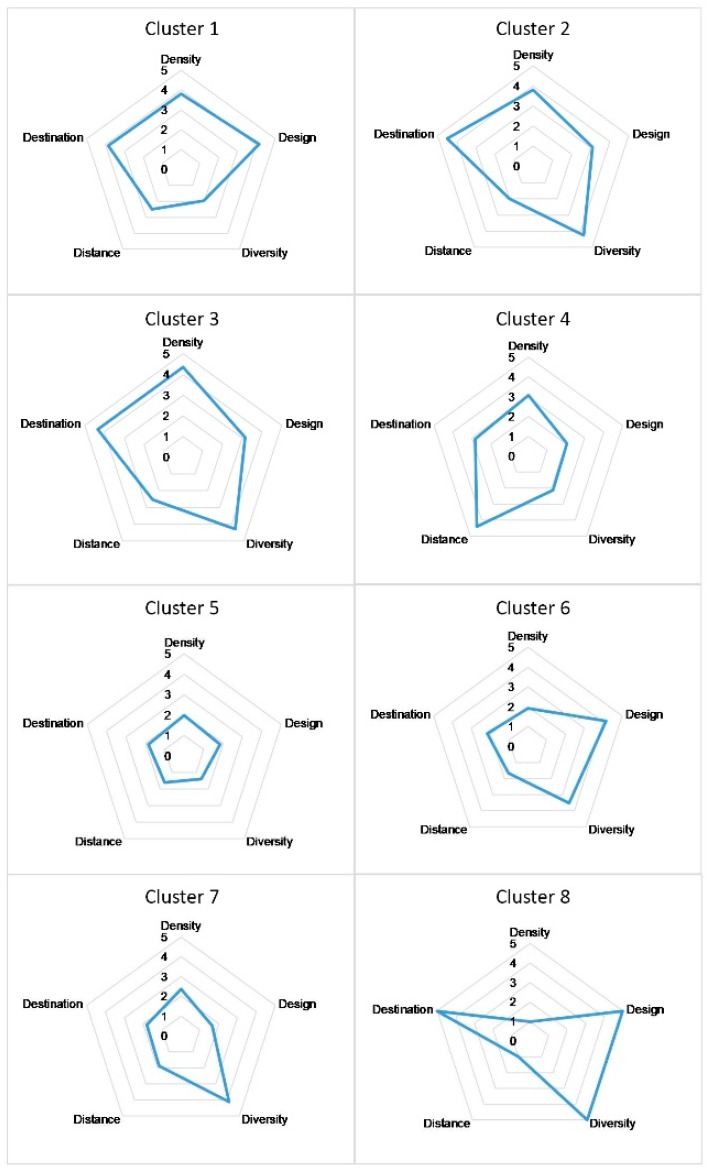
Radar charts displaying the average score for each variable by group.

**Table 1 ijerph-16-01782-t001:** Distribution of streets with different quality levels.

Type A: Clusters 1, 2, 3 (558 of 1231, 45%), High Quality	Type B: Cluster 4 (168 of 1231, 14%), Medium Quality	Type C: Clusters 5, 6, 7 (505 of 1231, 41%), Poor Quality
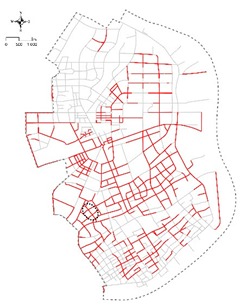	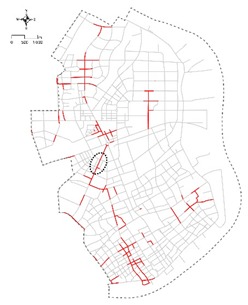	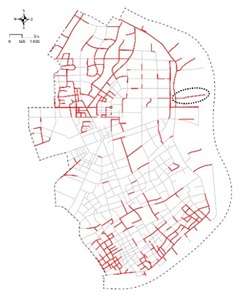
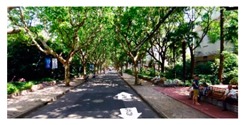	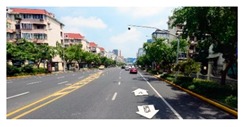	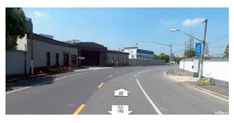

**Table 2 ijerph-16-01782-t002:** Inter-rater agreement of “machine” and “experts.”

Agreement	Expected Agreement	Weighted Kappa	Std. Err.	Z	Prob > Z
84.17%	54.61%	0.6512	0.0988	6.59	0.0000

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
