# Peer review of "A Systematic Measurement of Street Quality through Multi-Sourced Urban Data: A Human-Oriented Analysis"

_ijerph, 2019, doi:10.3390/ijerph16101782_

Round 1
Reviewer 1 Report
Type of manuscript: Article
Title: A systematic measurement of street quality through multi-sourced urban
data: a human-oriented analysis
General comments
The article is novel in reinterpreting the 5D using
machine learning, crowd sourcing data and network science techniques.
The breadth is impressive as many of the sections can be individual
papers.
I totally agree the method is useful to encapsulates the 5D into a single measurement which can be useful for practice and policies.
My main concern is that the section for validation is weak when
compare to the other sections.
Validation here is important as it will
make your index more convincing.
It might be useful to see how previous
5D studies conduct its validation and to give more details and summaries
on how the experiments were conducted.
Minor comments
Density
Temporal density is interesting and probably deserves its own paper with more details.
Design
Similarly, there could also be more details. For example,
Was the segnet a pretrained model or was the model trained specifically for this context? Please clarify in text.
Why was an ANN used in predicting quality preferences? How many hidden layers were used in the ANN? Please clarify in text.
Diversity
I assume SW is the entropy formula(nlogn) but for clarity please define SW more explicitly (formula).
Transit
It would be useful to define this more explicitly (formula). Are there any radius to the distance metric?
5D
Please provide descriptive statistics for the 5D after the maps.
Clustering analysis
For the dendrogram, please label the eight clusters on the branch.
Group 8 is indeed unique and might be worth checking in more detail.
Author Response
General comments
The article is novel in reinterpreting the 5D using machine learning, crowd sourcing data and network science techniques.
The breadth is impressive as many of the sections can be individual papers.
I totally agree the method is useful to encapsulates the 5D into a single measurement which can be useful for practice and policies.
My main concern is that the section for validation is weak when compare to the other sections.
Validation here is important as it will make your index more convincing.
It might be useful to see how previous 5D studies conduct its validation and to give more details and summaries on how the experiments were conducted.
Response:
Thank you for the comments!
To achieve a more subjective reliability validation, we adopted kappa statistics (using STATA) in revised section 4.3 (page 16, line 466 - 471) to measure the inter-rater reliability.
Literature review has been expanded on previous 5D studies (References [5-12]) and classic urban design literature [23, 24,59, 60, 61, 62, 63] for better summarizing knowledge gap.
Minor comments
Density
Temporal density is interesting and probably deserves its own paper with more details.
Response:
Thanks a lot for your kind suggestion. We agree that the temporal density is quite important. Nevertheless, the current manuscript is already quite long (9,300 words). Thus, we prefer to claim the importance of temporal density and address this issue systematically in our following papers (section 5.4, page 17, line 521-523).
Design
Similarly, there could also be more details. For example,
Was the segnet a pretrained model or was the model trained specifically for this context? Please clarify in text.
Why was an ANN used in predicting quality preferences? How many hidden layers were used in the ANN? Please clarify in text.
Response:
The SegNet applied herein is a pre-trained model provided by researchers from the University of Cambridge. We did not re-train it for this specific context as an empirical study utilizing this algorithm in Chinese cities performed well. There is only one hidden layer used in the ANN. It has been clarified in the revised manuscript (section 3.3.2, page 6, line 270-276).
Diversity
I assume SW is the entropy formula(nlogn) but for clarity please define SW more explicitly (formula).
Response:
Reference of Shannon-Winner index (reference [70]), details of POI data, and the formula of SW have been added into the revised manuscript (section 3.3.3, page 11, line 330 - 342).
Transit
It would be useful to define this more explicitly (formula). Are there any radius to the distance metric?
Response:
Thanks for pointing it out. The radius to the distance metric is 1km (15 min walking distance), revised in section 3.3.5 (page 11, line 361-363).
5D
Please provide descriptive statistics for the 5D after the maps.
Response:
Thank you very much for your suggestion! The descriptive statistics of individual variables have been added as box plots (figure 11) in section 4.1 (page 11-12, line 366-374).
Clustering analysis
For the dendrogram, please label the eight clusters on the branch.
Group 8 is indeed unique and might be worth checking in more detail.
Response:
Thanks for raising this concern! The cluster label, specific clustering algorithm has been added into the manuscript. The dendrogram summarizes eight clusters of streets with cluster 8 has only one street. Due to the large sample size (N=1,231), we focus on the seven clusters to see similarities and differences between them.
We highly appreciate the reviewer taking the time to offer us comments and insights related to the paper. We would be glad to respond to any further questions and comments that you may have.

Reviewer 2 Report
Please see the attached file for my review comments.

Author Response
Comment 1:
First of all, this paper needs a better justification for why a big-data driven analysis is more human-oriented. How are the GPS-based LBS data more “human-oriented” than conventional residential density or employment density? How are ANN-based scores more “human-oriented” than visual preference surveys?
Response:
LBS are location that are run and visited by people. Their spatial distribution at street link level is spatially much closer to the daily routine and experience of people while residential and population density are area based summary that abstract such daily pattern to an arbitrary area (MAUP ). The LBS map shows subtle differentiation in pattern from street to street within their street configuration. This is not captured by area-based statistics. ANN based score is systematisation of visual surveys – encoder and decoder – all link streets, every 20 m, bi-directional are decoded for an overall set of features that are semantically segmented from each image. In themselves, there are not expressing human preference yet in combination with other dimensions by:
1. association with LBS, street layout accessibility, proximity to transit etc. – people voting with their feet – although it is possible that well-used places have poor visual qualities because these places are mandatory activities – yet given the broad weekday and weekend day sampling it is very likely that they place have also an appeal to the population frequenting them
2. validation from visual guideline established according to visual surveys and corroborated by experts.
The revised manuscript addressed these issues by summarizing Literature review and knowledge gap in Section 2 (page 4-5, line 191-208), clarifying how ANN applied (section 3.3.2, page 8-9, line 278-283, line 293 to 301), and adding more details of input data (section 3 & section 4.1).
Comment 2:
As one of the goals of this paper is to develop street quality measurement with new data, I believe that the process and software used need to be described in more details. For example, how do the real-time LBS data from Tencent look like? How did the author use SegNet (some capture images to describe the process would be helpful)? How are PoIs classified into different urban functions? Especially, when computing diversity, the authors identified “the total PoIs within a walkable distance of a street segment” (p.9). I wonder what (and how many) types of facilities are exactly included. What are the “different urban functions” (p.9) used in the entropy calculation? Readers would want to know more details.
Response:
Thanks for the suggestion! Some details of the variables have been added in the revised manuscript, especially in section 3.3.2 (page 8-9, line 278-283, line 293 to 301) and section 3.3.3 (page 11, line 330 -342) to answer reviewer’s question. Figure 3 (Sample of LBS data from Tencent), Figure 5 (Distribution of SVIs collecting points), Figure 6(a schematic architecture of SegNet), Figure 10 (Distribution on POIs) were also added for better understanding.
Comment 3:
Even though the cluster analysis results are validated qualitatively, some quantitative measures need to be also explored. The authors should describe which clustering algorithm was used for what reasons (e.g., single linkage, average linkage, Ward). Also, because a cluster analysis is unsupervised and any number of clusters is possible, some statistics exist to determine the optimal number of clusters, such as Calinski and Harabasz index and Silhouette index. If the authors’ goal was to classify the street segments into three groups—from poor to high quality, they could have used k-means clustering (or a similar one) instead of hierarchical clustering (plus a subsequent qualitative classification from eight to three).
Response:
Thank you very much for raising this concern! The specific clustering algorithm has been added into the manuscript.
Although a fixed number of clusters is possible to be determined by using the statistics suggested by the reviewer or simply using an “elbow” method. However, the ‘best’ number of clusters is not always appropriate especially in a study with a large sample size (N=1,231, the ‘best’ number of clusters would be 20-40 in this study). Since the determination of this number is not an aim of this study, we chose to use hierarchy cluster, the number of groups need not be known a priori.
We agree that there is a confusion in the part of classifying 8 groups into 3 types based only on radar charts. Therefore, in the revised version, we re-classify the 8 clusters by taking both the tree structure and radar charts into account (section 4.2, page 14, line 418 -433). The new three types are more consistent with clusters, and also show a better agreement with experts’ evaluation.
Comment 4:
Lack of reliability test for the image classification for Design factor is another limitation of this research. An interrater reliability measure (e.g., kappa statistics) may help readers believe that the subjective “design” measures from experts are reliable.
Response:
Thank you for the suggestion! To achieve a more subjective reliability evaluation, we adopted kappa statistics (using STATA) in revised section 4.3 (page 16, line 466 - 471) to measure the inter-rater reliability.
Comment 5:
There is more literature to be reviewed regarding “how streetscape features affect the quality of life, and the following effects on physical activities and public health.” (p.1)
- Ameli, S. H., Hamidi, S., Garfinkel-Castro, A., & Ewing, R. (2015). Do better urban design qualities lead to more walking in Salt Lake City, Utah? Journal of Urban Design, 20(3), 393–410.
- Bahrainy, H., & Khosravi, H. (2013). The impact of urban design features and qualities on walkability and health in under-construction environments: The case of Hashtgerd New Town in Iran. Cities, 31, 17–28.
- Ewing, R., Hajrasouliha, A., Neckerman, K. M., Purciel-Hill, M., & Greene, W. (2016). Streetscape features related to pedestrian activity. Journal of Planning Education and Research, 36(1), 5–15.
- Park, K., Ewing, R., Sabouri, S., & Larsen, J. (2019). Street life and the built environment in an auto-oriented US region. Cities. 243-251.
- Rodríguez, D. A., Brisson, E. M., & Estupiñán, N. (2009). The relationship between segment-level built environment attributes and pedestrian activity around Bogota's BRT stations. - Transportation Research Part D: Transport and Environment, 14(7), 470–478. Rodríguez, Brisson, & Estupiñán, 2009
Response:
Thank you for offering the references! The revised paper has expanded literature review. (References [5-12], [59-63]). Knowledge gap and main contributions of this study were also summarized at the end of Section 2 (page 4-5, line 191-208).
We highly appreciate the reviewer taking the time to offer us comments and insights related to the paper. We would be glad to respond to any further questions and comments that you may have.

Reviewer 3 Report
1. The street is a vital part of the urban transportation vehicle network. It defines the capacity of the network and the travel costs for travelers. Besides the neighborhood characteristics and the physical characteristics of streets, the characteristics of networks are more direct and more human-oriented factors to the travel behavior and the route choice, e.g., the stochasticity, the environmental costs, etc, especially in the networks where electric vehicles are becoming popular. This is because these factors directly affect the interests of the travelers. More references are suggested to be reviewed.
2. Explain each abbreviation the first time you use it, e.g., support vector machine (SVM).
3. The authors claim that a systematic, multi-factor quantitative approach for measuring street quality with the support of multi-sourced urban data is proposed. However, according to the literature review of this study, such approaches have already been proposed. What is the difference between this study and the literature?
4. Since this article is a case study, the input data should be exposed in more detail.
5. The hierarchical cluster analysis cannot guarantee an optimum solution, thus evaluating the quality of the solution is necessary. However, the evaluation in Section 4.3 may be a little too subjective. Is an objective approach available for the evaluating?
6. The language must be improved. A native speaker may be helpful.
Author Response
Comment 1:
The street is a vital part of the urban transportation vehicle network. It defines the capacity of the network and the travel costs for travelers. Besides the neighborhood characteristics and the physical characteristics of streets, the characteristics of networks are more direct and more human-oriented factors to the travel behavior and the route choice, e.g., the stochasticity, the environmental costs, etc, especially in the networks where electric vehicles are becoming popular. This is because these factors directly affect the interests of the travelers. More references are suggested to be reviewed.
Response:
Thanks for the suggestion! The revised paper has expanded literature review. (References [5-12], [59-63]). Knowledge gap and main contributions of this study were also summarized at the end of Section 2 (page 4-5, line 191-208).
Comment 2:
Explain each abbreviation the first time you use it, e.g., support vector machine (SVM).
Response:
Thanks for the comments! Abbreviations were checked.
Comment 3:
The authors claim that a systematic, multi-factor quantitative approach for measuring street quality with the support of multi-sourced urban data is proposed. However, according to the literature review of this study, such approaches have already been proposed. What is the difference between this study and the literature?
Response:
Thank you for the comments! The revised paper addressed these issues by summarizing knowledge gap and main contributions of this study in the end of Section 2 (page 4-5, line 191-208). The main contributions of this study are:
• use of the 5Ds framework discussed above considering the distinctive physical and cultural features of the Asian cities context. This study strives to re-integrate these five variables and examine them in the Shanghai context with the primary objective of providing a comprehensive evaluation framework
• Use higher data resolution to measure street quality and their relation to physical activities from people’s daily behaviors, portraying a more human-oriented approach.
• Considering that intersection density cannot fully describe street layout configuration, and the relationship between part and whole for pedestrian and the serial view experience of the pedestrian, this paper uses a description of street layout and network science to present a more realistic pedestrian path choice routing analysis.
Comment 4:
Since this article is a case study, the input data should be exposed in more detail.
Response:
Thanks for the suggestion! The input data has been explained in more detail:
· Some details of the variables have been added in the revised manuscript, section 3.
· Figure 3 (Sample of LBS data from Tencent), Figure 5 (Distribution of SVIs collecting points), Figure 6(a schematic architecture of SegNet), Figure 10 (Distribution on POIs) were added for readers’ better understanding.
· As mentioned above, the descriptive statistics of individual variables have been added in section 4.1 (page 11-12, line 366-374).
Comment 5:
The hierarchical cluster analysis cannot guarantee an optimum solution, thus evaluating the quality of the solution is necessary. However, the evaluation in Section 4.3 may be a little too subjective. Is an objective approach available for the evaluating?
Response:
Thank you for the suggestion! To provide an objective approach for measuring the inter-rater reliability, a kappa value (using STATA) was utilized in revised section 4.3 (page 16, line 466 - 471) to measure the inter-rater reliability.
Comment 6:
The language must be improved. A native speaker may be helpful.
Response:
This manuscript has gone through a language proofreading by Elsevier Language Editing services before we submitted it toward IJERPH. Nevertheless, we agree that another round of language editing may be helpful. Thus, a native English speaker has reviewed and improved our revised manuscript. We hope the language quality of this paper would be better now. It would be our pleasure to make another round of language editing if the reviewer feels it is necessary.
We highly appreciate the reviewer taking the time to offer us comments and insights related to the paper. We would be glad to respond to any further questions and comments that you may have.

Round 2
Reviewer 1 Report
I believe my comments are well reflected and the manuscript has been improved substantially.
I still have some comments below.
Hierarchical clustering
a. Fig11 is useful as it shows the measures are highly skewed. Hierchical method typically finds spherical clusters
which might not be appropriate for highly skewed data. As a result, I recommend noramlising the skewed values in two stages. First: log(x) or sqrt(x), second: min-max(x) before re-applying the hierarchical clustering procedures.
b. Fig 13 is improved now. You might want to put your second clusters (type-A, type-B, type-C) with a second cut-off line.
c. I would consider including some internal cluster validation statistic such as silhouette plots, etc.
d. The external cluster validation results improved by including the Kappa Statistics in table 2.
As it is an important part of the research, I would put the
results of the external validation into a small summary table that summarises
the 5D expert ratings either in the main text or in the appendix.
Author Response
Reviewer 1
I believe my comments are well reflected and the manuscript has been improved substantially.
I still have some comments below.
Hierarchical clustering
a. Fig11 is useful as it shows the measures are highly skewed. Hierchical method typically finds spherical clusters which might not be appropriate for highly skewed data. As a result, I recommend noramlising the skewed values in two stages. First: log(x) or sqrt(x), second: min-max(x) before re-applying the hierarchical clustering procedures.
Response: Box plots illustrate that the measures are highly skewed. Therefore, before applying the hierarchical clustering procedures, we have ranked each variable and then divided into five equal quintiles accordingly, then gave a score of 1 to 5 (line 423-426). This simplified scoring system also allows the scores comparable to the evaluation from experts.
b. Fig 13 is improved now. You might want to put your second clusters (type-A, type-B, type-C) with a second cut-off line.
Response: The second cut-off line has been added into the revised paper - Figure 13.
c. I would consider including some internal cluster validation statistic such as silhouette plots, etc.
Response: The internal cluster validation statistics such as silhouette index are usually used to determine the optimal number of clusters. However, a ‘best’ number of clusters is not always appropriate especially in the study with a large sample size. In this study with N=1,231, the ‘best’ number of clusters would be larger than 20-40, which works from the mathematical viewpoint but it would be hard to find appropriate correspondences in urban planning and design practices. We guess it would be better to use current results which is easy to explain and understand by urban planners and designers. Considering the determination of this number is not the aim of this study, we did not add silhouette plots to avoid confusion. We could add it in the next round revision in if the reviewer insists.
d. The external cluster validation results improved by including the Kappa Statistics in table 2. As it is an important part of the research, I would put the results of the external validation into a small summary table that summarises the 5D expert ratings either in the main text or in the appendix.
Response: The data has been summarized as a table and has been provided as Appendix A.

Reviewer 2 Report
I believe this version reflects the reviewers’ comments well, and the three final clusters are easier to interpret and apply.
Two minor comments are:
Figure 6 or 7: please consider adding legend of streetscape elements (e.g., sky view, greenery, building façade, etc.)
It’s still difficult to understand how the ‘design score’ was computed. What did ANN exactly use as input data from the SegNet outcomes? Relative area of each of the six spatial elements? Related to this, please expand this explanation further: “there was not a clear linear relationship between the perceived quality and these key spatial elements” (page 9). Exactly, what aspects of the spatial elements were related to the perceived quality scores?
Author Response
Reviewer 2
I believe this version reflects the reviewers’ comments well, and the three final clusters are easier to interpret and apply.
Two minor comments are:
Figure 6 or 7: please consider adding legend of streetscape elements (e.g., sky view, greenery, building façade, etc.)
Response: Legend of six elements has been added into both figures.
It’s still difficult to understand how the ‘design score’ was computed. What did ANN exactly use as input data from the SegNet outcomes?
Response: The process of ANN algorithm has been explained in the revised manuscript (Line 335-345).
Relative area of each of the six spatial elements? Related to this, please expand this explanation further: “there was not a clear linear relationship between the perceived quality and these key spatial elements” (page 9).
Response: The reason “why there was no clear linear realationship” and “why a linear regression cannot address this complex and interacted relationship” has been added in the revised paper (line 324-334).
Exactly, what aspects of the spatial elements were related to the perceived quality scores?
Response: The six key elements related to the perceived quality scores were selected based on classical design theories from Jacobs [33] to Trancik [56] to Katz [57] and Montgomery [58], this part has been expanded In section 2.2 line 175-188.

Reviewer 3 Report
I can see that the authors made great efforts to this version. However, the authors added more references to the neighborhood characteristics and the physical characteristics of networks. I believe there must be some misunderstanding in Comment 1.
Different from the neighborhood characteristics and the physical characteristics of streets, the characteristics of networks is a combination of the physical and the behavioral characteristics. For example, the stochasticity and the environmental costs. Specifically, in the urban traffic network, the stochasticity comes from the different perception abilities of the travelers. The environmental cost is an internal cost perceived by travelers. Both of them are highly human-oriented. Moreover, to the best of our knowledge, Shanghai is the fastest growing city in electric vehicle ownership. In this context, the stochasticity and the environmental costs of a network may be more direct and human-oriented factors in the future. Thus, more references on the stochastic networks, environmental costs, and electric vehicles are recommended.
Since this paper is a case study, data exposure is critical. I suggest that the authors should publish their data to any public platform.
All other my concerned have been addressed.
Author Response
Reviewer 3
I can see that the authors made great efforts to this version. However, the authors added more references to the neighborhood characteristics and the physical characteristics of networks. I believe there must be some misunderstanding in Comment 1.
Different from the neighborhood characteristics and the physical characteristics of streets, the characteristics of networks is a combination of the physical and the behavioral characteristics. For example, the stochasticity and the environmental costs. Specifically, in the urban traffic network, the stochasticity comes from the different perception abilities of the travelers. The environmental cost is an internal cost perceived by travelers. Both of them are highly human-oriented. Moreover, to the best of our knowledge, Shanghai is the fastest growing city in electric vehicle ownership. In this context, the stochasticity and the environmental costs of a network may be more direct and human-oriented factors in the future. Thus, more references on the stochastic networks, environmental costs, and electric vehicles are recommended.
Response: We agree that the characteristics of networks is a combination of the physical and the behavioral characteristics. Therefore, apart from physical characteristics we have addressed in the original paper, the revised paper expanded on references on the effects of network characteristics on people’s travel walking and route choice, i.e. behavioural characteristics, especially in the context of Shanghai (section 1.2, line 60-71; section 2.2, line 208-212).
The main purpose of this paper is to develop a 5Ds analytical framework and further encourage physical activity and public health. Even though we believe that the electric vehicle share keeps increasing in Shanghai, and it is interesting to consider how these changes may effect environmental costs, we think this is beyond the scope of the present study, but it is an important topic for future studies.
Since this paper is a case study, data exposure is critical. I suggest that the authors should publish their data to any public platform.
Response: We have uploaded the datasets used in this paper.
1. LBS_Kernel.gdb which include LBS points.
2. database.gdb which includs following feature classes:
· poi_yangpu: POI related data
· yangpu_distrct: boundary of Yangpu District.
· Shanghai_WGS84_Road_sDNA_clip: a street path-center line map of Yangpu district extracted from route data in Baidu Maps (with accessibility result computed using the whole Shanghai map).
· Yangpu_Streetquality_variables: street quality results used in Figure 12.
All other my concerned have been addressed.
